

# Does ankle push-off correct for errors in anterior–posterior foot placement relative to center-of-mass states?

Jian Jin[1], Jaap H. van Dieën[1], Dinant Kistemaker[1], Andreas Daffertshofer[1,2] and Sjoerd M. Bruijn[1,2]

[1] Department of Human Movement Sciences, Faculty of Behavioural and Movement Sciences, Vrije Universiteit Amsterdam, Amsterdam, The Netherlands
[2] Institute of Brain and Behavior Amsterdam, Amsterdam, The Netherlands

## ABSTRACT

Understanding the mechanisms humans use to stabilize walking is vital for predicting falls in elderly. Modeling studies identified two potential mechanisms to stabilize gait in the anterior-posterior direction: foot placement control and ankle push-off control: foot placement depends on position and velocity of the center-of-mass (CoM) and push-off covaries with deviations between actual and predicted CoM trajectories. While both control mechanisms have been reported in humans, it is unknown whether especially the latter one is employed in unperturbed steady-state walking. Based on the finding of Wang and Srinivasan that foot placement deviates in the same direction as the CoM states in the preceding swing phase, and assuming that this covariance serves the role of stabilizing gait, the covariance between the CoM states and foot placement can be seen as a measure of foot placement accuracy. We subsequently interpreted the residual variance in foot placement from a linear regression model as "errors" that must be compensated, and investigated whether these foot placement errors were correlated to push-off kinetic time series of the subsequent double stance phase. We found ankle push-off torque to be correlated to the foot placement errors in 30 participants when walking at normal and slow speeds, with peak correlations over the double stance phase up to 0.39. Our study suggests that humans use a push-off strategy for correcting foot placement errors in steady-state walking.

## INTRODUCTION

Walking is a seemingly simple task for most of us. Yet, it involves intricate feedback control and an ongoing integration of sensory inputs and generation of motor outputs (*Rossignol, Dubuc & Gossard, 2006*; *Warren et al., 2001*; *Zehr & Stein, 1999*). A solid understanding of the mechanisms that underlie the stabilization of walking is crucial for identifying causes of falls in elderly (*McGibbon, 2003*). *In-silico* studies singled out basic principles of locomotion. Two examples are feedback control of foot placement and of ankle push-off. Both of these mechanisms have been shown to stabilize comparably simple walking models in the presence of perturbations. For instance, *Hobbelen & Wisse (2008)* illustrated that modulating ankle push-off torque *via* feedback from the leading leg's angle substantially

Corresponding author
Sjoerd M. Bruijn, s.m.bruijn@vu.nl

improved robustness against perturbations in a sagittal plane flat-feet walker. According to *Byl & Tedrake (2008)*, feedback-controlled foot placement with a constant push-off magnitude appears more robust than merely modulating the push-off magnitude in a point-feet walker model. In addition, in inverted pendulum models, a combined foot placement and push-off control based on feedback from mid-stance speed demonstrated good robustness (*Bhounsule, 2015*; *Kelly & Ruina, 2015*; *Zaytsev, Wolfslag & Ruina, 2018*). Also in human walking, one can find ample evidence that foot placement is based on CoM position and/or velocity. This applies to both steady-state (*Redfern & Schumann, 1994*; *Wang & Srinivasan, 2014*) and perturbed walking (*Joshi & Srinivasan, 2019*; *Vlutters, van Asseldonk & vander Kooij, 2018b*). Moreover, after a perturbation, ankle push-off seems to modulate with deviations of CoM trajectories from the desired trajectories (*Afschrift, de Groote & Jonkers, 2021*; *Fettrow et al., 2019*; *van Mierlo et al., 2021*; *Vlutters, van Asseldonk & vander Kooij, 2016*). Thus far this evidence is restricted to perturbed walking, but we here subsume that humans also use a feedback-controlled push-off strategy to stabilize steady-state walking.

In human walking, foot placement along the anterior-posterior (AP) direction is correlated to CoM states, *i.e.,* CoM position and velocity. *Wang & Srinivasan (2014)* showed that CoM states at mid-stance and heel strike predict about 33% of the variance in AP foot placement. Recently, *Liu (2021)* replicated this finding in steady-state walking and additionally in perturbed walking. This study by Liu revealed that a different foot placement model is needed to describe responses to slip-like and trip-like treadmill perturbations, suggesting that the control of foot placement may be different between steady-state and perturbed walking. By the same token, ankle push-off during the double stance phase seems to modulate with AP pelvis perturbations: *van Mierlo et al. (2021)* found adjustments of the double stance duration and the AP center-of-pressure (CoP) trajectory to correct perturbed CoM states towards desired states at the end of the double stance phase; and, *Afschrift, de Groote & Jonkers (2021)* demonstrated that for AP pelvis perturbations and for treadmill belt speed perturbations, the deviations in CoM states from the steady-state trajectory accurately predicted the ankle moment after a neural delay of about 100 ms. In steady-state walking, the percentage of explained variance of AP foot placement from the CoM states increased from 33% to 70% during the double stance phase (*Wang & Srinivasan, 2014*). That is, there may be modulations of the CoM states through ankle push-off control.

Since CoM states covary with AP foot placement, either one does not suffice as the input for push-off control. Instead, more appropriate feedback signals would integrate both terms, representing the covariance between the CoM states and foot placement, or *foot placement error*, first introduced by *van Leeuwen et al. (2021)*. It is defined as the difference between the actual foot placement and the ideal foot placement based on the CoM states at heel strike. This can be estimated by linearly regressing CoM states with foot placement (*Wang & Srinivasan, 2014*). Maintaining the covariance between the CoM states and foot placement has been suggested to be a useful control strategy for gait stability both in simple walking models (*Hof, 2008*; *Joshi & Srinivasan, 2019*; *Maki & McIlroy, 1999*; *Townsend, 1985*) and in humans (*Joshi & Srinivasan, 2019*; *Redfern & Schumann, 1994*; *Verrel, Lövdén & Lindenberger, 2010*; *Wang & Srinivasan, 2014*). Assuming the presence of foot placement
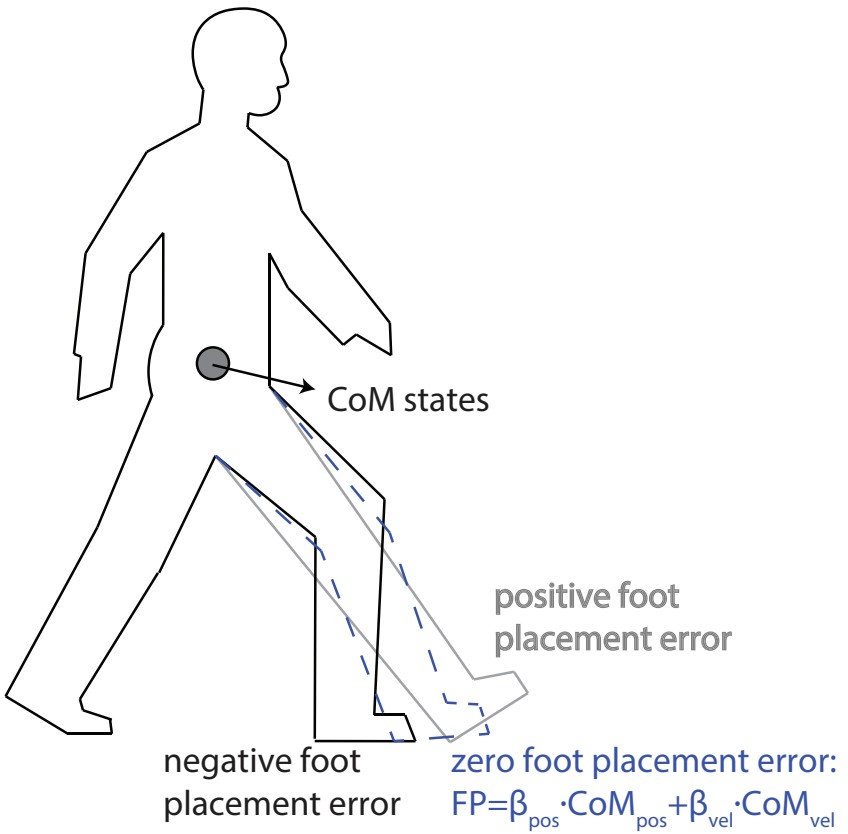

**CoM states**

positive foot
placement error

negative foot
placement error

zero foot placement error:
$FP = \beta_{pos} \cdot CoM_{pos} + \beta_{vel} \cdot CoM_{vel}$

**Figure 1 Illustration of positive and negative foot placement errors.** For certain CoM states (CoM AP position and velocity at heel strike), a too large step yields a positive foot placement error, while a too small step leads to a negative foot placement error. A vanishing foot placement error implies linear predictions of foot placement from CoM states.

control, residual variance in foot placement from a linear regression model can be viewed as foot placement *errors* that must be corrected.

As shown in Fig. 1, a positive foot placement error in the AP direction indicates that foot is placed more forward with respect to a linear prediction from the CoM states. In this case, there is a higher collision loss due to larger step length (*Adamczyk & Kuo, 2009*), and the walker's post-collision total energy will be lower, requiring a stronger push-off from the trailing leg and/or from the hip extension torque *via* the leading leg to compensate for this larger collision loss. Here, we assumed the collision loss to be balanced by the push-off work on the CoM during the double stance phase, as observed in human walking (*Kim & Park, 2012*). We hence expected the push-off force/torque magnitude to be proportional to the AP foot placement error, *i.e.,* a stronger push-off for larger positive foot placement errors, and a weaker push-off for more negative foot placement errors.

We examined push-off modulations by investigating how kinetic time series were modulated during the double stance phase as a function of the foot placement error. Next, we evaluated the correlations between foot placement errors and three kinetic time series: the combined AP ground reaction force (GRF), the trailing leg's AP GRF, and the trailing

leg's ankle moment. The kinetic time series were chosen instead of the (scalar) kinematic variables like work or impulse because the kinetic pattern could serve to illuminate the mechanisms related to the correction of foot placement errors. We did not select ankle power as outcome measure mainly because ankle power is determined by ankle moment and angular velocity, and modeling studies (*Bhounsule, 2015*; *Hobbelen & Wisse, 2008*; *Kelly & Ruina, 2015*; *Zaytsev, Wolfslag & Ruina, 2018*) suggested that modulating the magnitude of ankle push-off moment without joint angular velocity input was sufficient for improving gait robustness. For every participant, we also compared the mean kinetics of ten most positive foot placement errors with the mean kinetics of ten most negative foot placement errors to study how the push-off may differ when correcting positive and negative foot placement errors. We selected the combined AP GRF because it represents the summed effect of ankle push-off by the trailing leg and braking by the leading leg. The use of the trailing leg's AP GRF was motivated by the idea that the CoM is mainly accelerated *via* the trailing leg's AP GRF (*Hernández et al., 2009*). The use of ankle moment was motivated by modeling studies that indicated ankle push-off to stabilize gait (*Hobbelen & Wisse, 2008*; *Kerimoglu et al., 2021*; *Kim & Collins, 2017*) and to provide the bulk of the trailing leg's propulsive force (*Franz & Kram, 2014*). This is due to a higher gain from ankle moments to the trailing leg's AP GRF compared to hip or knee moments (*Biewener et al., 2004*; *Toney & Chang, 2016*).

## METHODS

We used existing data from steady-state normal and slow treadmill walking experiments (*van Leeuwen et al., 2020a*; *van Leeuwen et al., 2020b*). Below, we briefly sketch data collection and processing but we refer to *van Leeuwen et al. (2020a)* for more details. The data can be found at https://doi.org/10.5281/zenodo.4229851 and the code can be found at https://zenodo.org/record/7393791#.Y43Gvy2ZOqQ.

### Participants

Thirty participants were included (19 female, 11 male, age 30 ± 7 years, weight 70 ± 13 kg, height 1.73 ± 0.08 m; mean ± sd). None of them reported injuries or balance issues that could affect their gait pattern. All participants signed informed consent before the experiment. Ethical approval (VCWE-2018-159) had been granted by the ethics board of the Faculty of Behavioural and Movement Sciences, Vrije Universiteit Amsterdam, prior to conducting the experiment.

### Protocol

Every participant walked on a treadmill at a constant belt speed of $v = 0.4\sqrt{gl}$ m/s (*normal* walking condition) and $v = 0.2\sqrt{gl}$ m/s (*slow* walking condition), with $g = 9.81$ m/s² being the gravitational constant and $l$ the leg length. A metronome served to impose stride frequency to minimize stride frequency variations. Participants were asked to match their right heel strikes to the beat. The imposed frequency during normal/slow trials was customized as the average preferred stride frequency during the last hundred steps of the familiarization trial without metronome at each speed. The normal walking trials

lasted five and the slow walking trials ten minutes each to ensure that data of at least 200 consecutive strides were collected. Normal and slow speed trials were randomized in order and separated by sufficient breaks to prevent fatigue.

## Data collection and processing

Participants walked on an instrumented dual-belt treadmill (Motek-Force-link, Amsterdam, Netherlands). Full-body kinematics were recorded using an active 3D motion analysis system and cluster markers on all segments (Optotrak, Northern Digital Inc, Waterloo ON, Canada). For every participant and condition, we analyzed the last 200 consecutive strides without data quality issues (*e.g.*, limited marker visibility, large noise, *etc*). GRF data were collected using a force plate integrated in the treadmill. Gait events (heel strike and toe-off) were detected based on the so-called "butterfly pattern" of the combined CoP trajectory. This method, validated by *Roerdink et al. (2008)*, has the advantage that it does not require the setting of an ad-hoc threshold. The trajectories of the segments and the kinematics were calculated using a 3D linked segmented model (*Kingma et al., 1996*) based on the coordinates of markers and anatomical landmarks. The kinetics, including ankle moments in the sagittal plane, was calculated from the measured GRFs and the lower body kinematics using bottom-up inverse dynamics (*Hof, 1992*; *Kingma et al., 1996*). For subsequent analysis, the kinematic and kinetic time series of a step/stride (defined from heel strike to contralateral/ipsilateral heel strike) were time-normalized. We accounted for the variations in timing of heel strike and subsequent push-off within a step[1] by segmenting the time window of a step to single-stance and double stance sub-windows with fixed time length per walking condition.

## Linear foot placement model

Similar to previous studies (*van Leeuwen et al., 2020a*; *Wang & Srinivasan, 2014*), we fitted a linear model between the CoM states and AP foot placement. This full body CoM state was derived from a weighted sum of the body segments' CoM, which in turn was estimated from the percent longitudinal distances of the body segments' CoM to neighboring bony landmarks, relative to the lengths of respective segments (*de Leva, 1996*). The predictor CoM states included the AP position (the horizontal distance from stance foot to CoM) and the AP velocity, *i.e.*, $\text{COM}_{\text{pos}}$ and $\text{COM}_{\text{vel}}$, respectively. The predictor and dependent variables were de-meaned prior to regression to ensure a zero intercept. The linear foot placement model reads as follows:

$$\text{FP}_i(j = 100\%) = \beta_{\text{pos}}(j) \cdot \text{CoM}_{\text{pos},i}(j) + \beta_{\text{vel}}(j) \cdot \text{CoM}_{\text{vel},i}(j) + \varepsilon_{\text{FP},i}(j), \tag{1}$$

which predicts AP foot placement at heel strike $\text{FP}_i(j = 100\%)$ based on CoM AP position $\text{CoM}_{\text{pos},i}(j)$ and velocity $\text{CoM}_{\text{vel},i}(j)$ at step $i$ and phase $j$. More specifically, $\text{FP}_i$ represents the (de-meaned) AP distance between the trailing and leading leg in step $i$, $\beta_{\text{pos}}(j)$ and $\beta_{\text{vel}}(j)$ are the phase-dependent regression coefficients obtained from the least square fit, and $\varepsilon_{\text{FP},i}(j)$ denotes the phase-dependent residual from the linear regression. The variable $j$ indicates the (normalized) phase in a step (0–100% from heel strike to contralateral heel strike). We used the phase instant at heel strike (100%) to compute the foot placement

FP$_i$ and the foot placement error $\varepsilon_{\text{FP},i}$, which we assume to be corrected by the subsequent push-off, see below.

## Correction of foot placement error

Using regression analysis, we evaluated how well the foot placement error at heel strike correlated to the combined AP GRF, the trailing leg's AP GRF, and the trailing leg's ankle moment during the double stance. The general model relating foot placement errors and the subsequent kinetic time series (indicated by $F$) was defined as:

$$F_{i+1}(j) = \beta_{\varepsilon_{\text{FP}}}(j) \cdot \varepsilon_{\text{FP},i}(j = 100\%) + \varepsilon_{i+1}(j), \tag{2}$$

where $\beta_{\varepsilon_{\text{FP}}}(j)$ is the phase-dependent linear regression coefficient, and $\varepsilon_{i+1}(j)$ is the residual of the new linear regression. The step index $i+1$ was used because we intended to investigate whether the foot placement error $\varepsilon_{\text{FP},i}$ at the end of previous step $i$ was corrected by the kinetic variables in the subsequent step $F_{i+1}$, in particular during the double stance phase. However, given that the ankle push-off moment could potentially begin before contralateral heel strike, and that the stored elastic energy during the preceding ankle plantarflexion is released during ankle push-off and influences the ankle extension moment, we also replaced step index $i+1$ by $i$ in Eq. (2) to investigate the kinetic time series earlier than the heel strike. As such, we included and analyzed the kinetic time series of a complete stride.

We selected three types of kinetic variables $F$ to correlate with foot placement errors $\varepsilon_{\text{FP}}$: (1) combined AP GRF, which represents the sum of anterior-posterior GRF at both legs, with larger positive values indicating a larger forward GRF; (2) trailing leg's AP GRF; (3) trailing leg's ankle moment, with positive values indicating an internal plantar-flexion moment. All position coordinates were defined with respect to the stance foot (trailing foot). As illustrated in Fig. 1, a larger positive foot placement error indicates a more forward foot placement location than the linear prediction from the CoM states. We expected positive correlations between foot placement errors and kinetic time series ($F$) during the subsequent double stance phase, *i.e.*, a larger combined/trailing leg's AP GRF or ankle moment for a larger positive foot placement error, and vice versa. We assume the resulting correlations to suggest feedback control. Yet, we have to admit that correlations may also arise from passive dynamics (*Patil, Dingwell & Cusumano, 2019*) or from intrinsic muscle properties (*e.g.*, preflexes stabilizing movements through nonlinear viscoelastic properties of muscles when in contact; cf. *Loeb, 1995*). We return to this in the Discussion section.

In addition to the regression analysis, we compared the kinetic time series of the "most positive" with that of the "most negative" foot placement errors. Specifically, the foot placement errors of each participant (200 strides) were sorted from low to high, and the ten most positive and ten most negative foot placement error strides were analyzed, by extracting the mean kinetic time series from these sets of ten strides.

## Statistics

All statistical tests and analysis were performed in Matlab (v2017b, The Mathworks Inc., Natick, MA), including the multi-linear regression of the foot placement model.

Correlation values are presented as group level means and individual data. The mean correlations were obtained by (1) first applying Fisher's r-to-z transform to guarantee the normality of the correlation coefficients; (2) then averaging the z-values over all participants per condition; and, (3), inverting the Fisher transform to obtain the group mean correlation coefficients per condition. We tested the correlation coefficients for significance using statistical parametric mapping (*Friston et al., 2007*) as implemented in the SPM1D toolbox (https://spm1d.org/). In brief, SPM1D allows for statistical testing of time series data, considering the interdependence (smoothness) between time samples. To test for differences between the kinetic time series of the "most positive" and "most negative" foot placement errors, we also performed an SPM1D paired $t$-test. The significance level of every test was set to $\alpha = 0.05$.

# RESULTS

The variance in AP foot placement explained by the CoM states ($R^2$) increased over the single stance phase (Fig. 2, left of the dashed vertical line) and reached values between 50% and 80% during the double stance phase in both normal and slow walking (Fig. 2, right of the dashed vertical line). Regression coefficients for CoM AP velocity were larger than for AP position (Fig. S1). The significance of the correlation coefficients for all 30 participants was also presented in Fig. S1, which showed that the linear foot placement model (Eq. (1)) reached significance in the majority phases from toe-off to contralateral toe-off for the majority of participants. Note that in order to demonstrate foot placement accuracy, a step here in Fig. 2 was defined from toe-off to contralateral toe-off instead of from heel strike to contralateral heel strike which was defined in Eqs. (1) and (2).

Although a large proportion of foot placement variance could be explained by the CoM states at heel strike, a fair amount of unexplained variance remained, *i.e.,* there were substantial foot placement errors. The distributions of foot placement errors across all participants for normal and slow walking were close to Gaussian with zero mean (Fig. 3A). Note that zero mean of the foot placement errors is the result of the de-meaned FP. The variance of foot placement errors in slow walking was larger than in normal walking (Fig. 3A). The group means of the 10 most positive and 10 most negative foot placement errors in slow walking had larger magnitudes compared to those in normal walking (Fig. 3B).

Across participants, the foot placement errors at heel strike were correlated with the combined AP GRF during the subsequent double stance phase in normal and slow walking (peak mean correlations over participants up to 0.34 and 0.28, respectively; see Fig. 4A). The significance of such correlation coefficients for all 30 participants was presented in Fig. S2, which showed that the correlation coefficients reached significance in early phases of the double stance phase for all except one participant in slow walking (participant 08). High positive correlations occurred in the first half of the double stance phase and decreased to negative correlations over the latter half of the double stance phase. Correlations were close to zero prior to foot placement, which implies that foot placement can hardly be predicted based on the combined AP GRF prior to foot placement. Correlations were also close to zero at foot placement. This suggests that preemptive push-off indicated by the combined

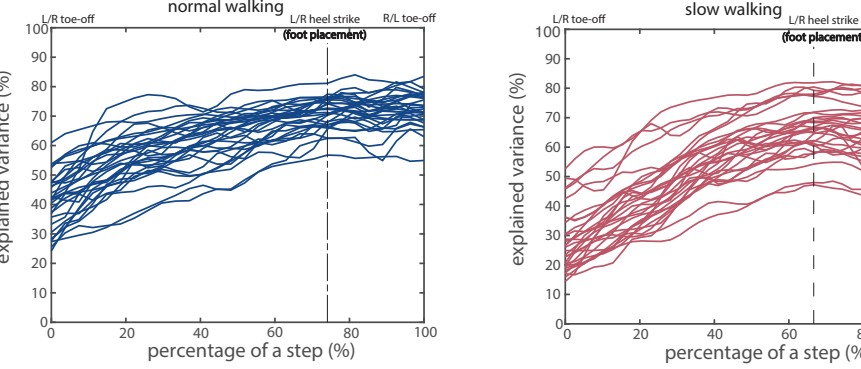

**Figure 2   Proportion of foot placement variance explained by CoM states, for normal (A) and slow walking (B).** Every curve represents the explained variance for a participant. The dashed vertical lines correspond to foot placement, and the phases after heel strike correspond to the double stance phase.

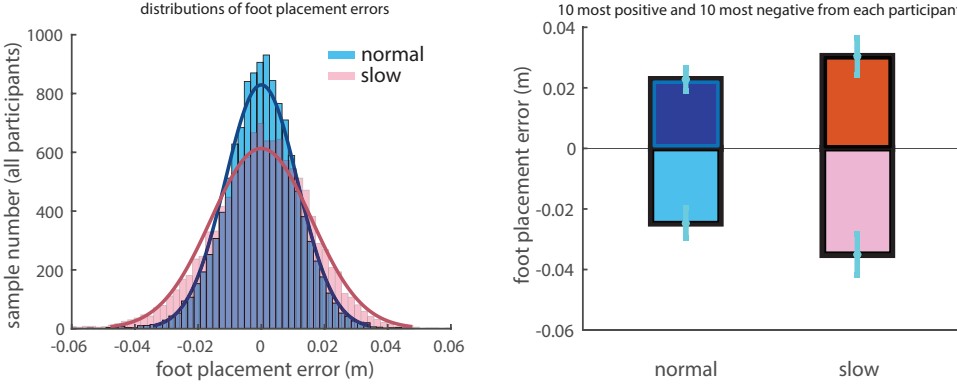

**Figure 3   Distributions of foot placement errors.** (A) Distributions of foot placement errors for normal and slow walking for all participants. Gaussian distributions are fitted to the data in two conditions. (B) The group mean values of the averaged 10 most positive and 10 most negative foot placement errors per participant. Error bars indicate the standard deviation across participants.

AP GRF did not cause foot placement errors. Correlations were significant and negative at the end of the stride cycle during normal walking (before the subsequent heel strike, mean correlations reached −0.43, see Fig. 4A and Fig. S2).

Likewise, the foot placement errors at heel strike were correlated with the trailing leg's AP GRF during the subsequent double stance phase in normal and in slow walking (peak mean correlations over participants up to 0.45 and 0.41, respectively; see Fig. 4B). The significance of such correlation coefficients for all 30 participants was presented in Fig. S3, which showed that the correlation coefficients reached significance around the middle of the double stance phase for all participants in normal and slow walking. Correlations were close to zero at the beginning and at the end of the double stance phase, suggesting that preemptive, early and late push-off did not influence foot placement errors.

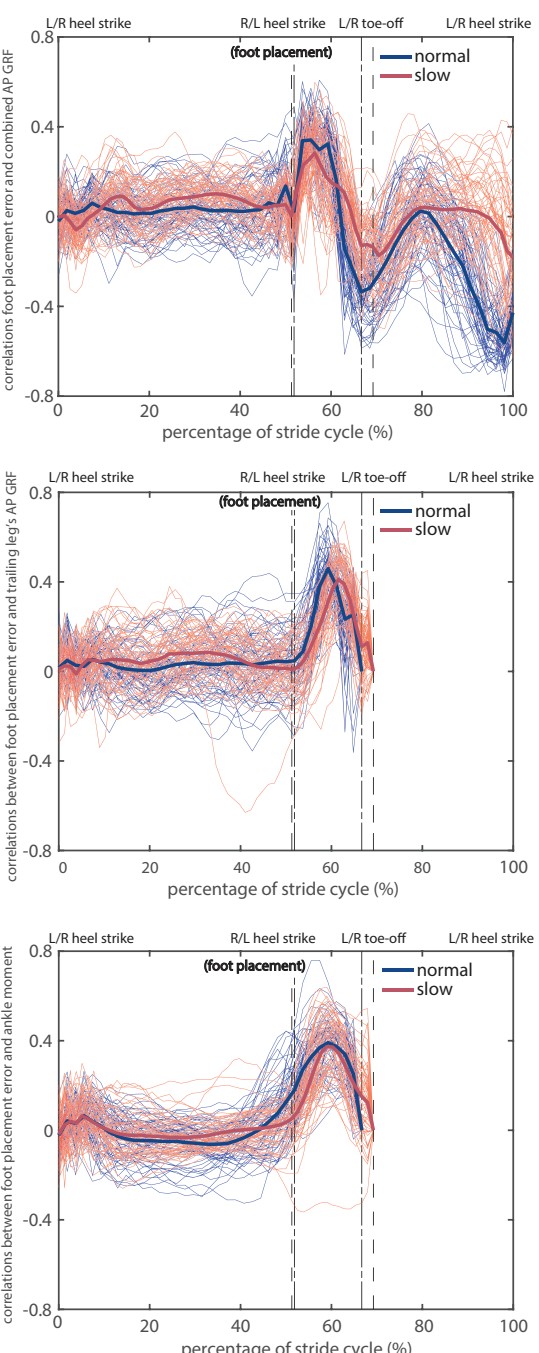

**Figure 4** **Correlations between foot placement errors at heel strike (left dash-dotted and dashed lines) and different kinetic time series in normal and slow walking.** (A) Combined AP GRF; (B) trailing leg's AP GRF; (C) ankle moment. The thin curves are correlations per participant and per foot, the thick ones are means over participants. The phases between the dash-dotted lines or the dashed lines correspond to the double stance phase in normal and slow walking, respectively. Note that the trailing leg's AP GRF and ankle moment were zero during the swing phases at the end of the stride cycle, thus the correlations were not shown.

The ankle moments during the double stance were correlated with foot placement errors at heel strike in normal and slow walking (peak mean correlations over participants up to 0.39 and 0.37, respectively; see Fig. 4C). Correlations with ankle moments were also highest in the middle of the double stance phase and were positive in the beginning of the double stance phase. The significance of such correlation coefficients for all 30 participants was presented in Fig. S4, which showed that the correlation coefficients reached significance around the middle of the double stance phase for all except two participants in slow walking (participant 14 and 35). The correlations during the double stance phase for participant 28 were negative, opposite to what we expected (Fig. 4C and Fig. S4).

In line with the correlations, all the kinetic time series during the double stance phase were larger for "most positive" foot placement errors than for "most negative" foot placement errors in both walking conditions (Figs. 5A–5C). As shown in Figs. 5A–5C, the combined AP GRF was higher in the first half and lower in the end of the double stance phase for "most positive" foot placement errors; the trailing leg's AP GRF and ankle moment were higher in the middle of the double stance phase for "most positive" foot placement errors. The differences of all three kinetic time series between "most positive" and "most negative" foot placement errors in both normal and slow walking reached significance in the majority of the double stance phase, see Fig. S5.

## DISCUSSION

We sought to test the hypothesis that humans employ ankle push-off control to compensate for errors in AP foot placement during steady-state walking. Our correlations between the foot placement errors at heel strike and all three kinetic time series during the subsequent double stance phase appear to confirm this. This does not come as a surprise, as several modeling studies demonstrated the capacity of ankle push-off modulations in improving robustness against perturbations (*Hobbelen & Wisse, 2008*; *Kim & Collins, 2013*; *Kim & Collins, 2017*; *Stephens & Atkeson, 2009*). Moreover, correlations between push-off modulations and deviations in CoM trajectories from AP perturbations have already been found in perturbed human walking (*Afschrift, de Groote & Jonkers, 2021*; *Rafiee & Kiemel, 2020*; *van Mierlo et al., 2021*; *Vlutters, van Asseldonk & vander Kooij, 2016*). Such correlations arguably arise from feedback mechanisms. Recently, *van Leeuwen et al. (2021)* reported that errors in medio-lateral foot placement were corrected by ankle moments during the subsequent stance phase. We here supplemented these findings by showing evidence for a push-off mechanism used to correct for errors in foot placement during steady-state walking in the AP direction.

In steady-state normal walking, among the three investigated kinetic time series, we found the largest peak (over the stride cycle) of the mean correlations (across all participants) to be the one between foot placement errors and trailing leg's AP GRF (up to 0.45), followed by (peak) correlations between the foot placement errors and trailing leg's ankle moment (up to 0.39), and the (peak) correlations between the foot placement errors and combined AP GRF (up to 0.34). Feedback control based on foot placement error may primarily serve for control of the push-off force from the trailing leg, which is

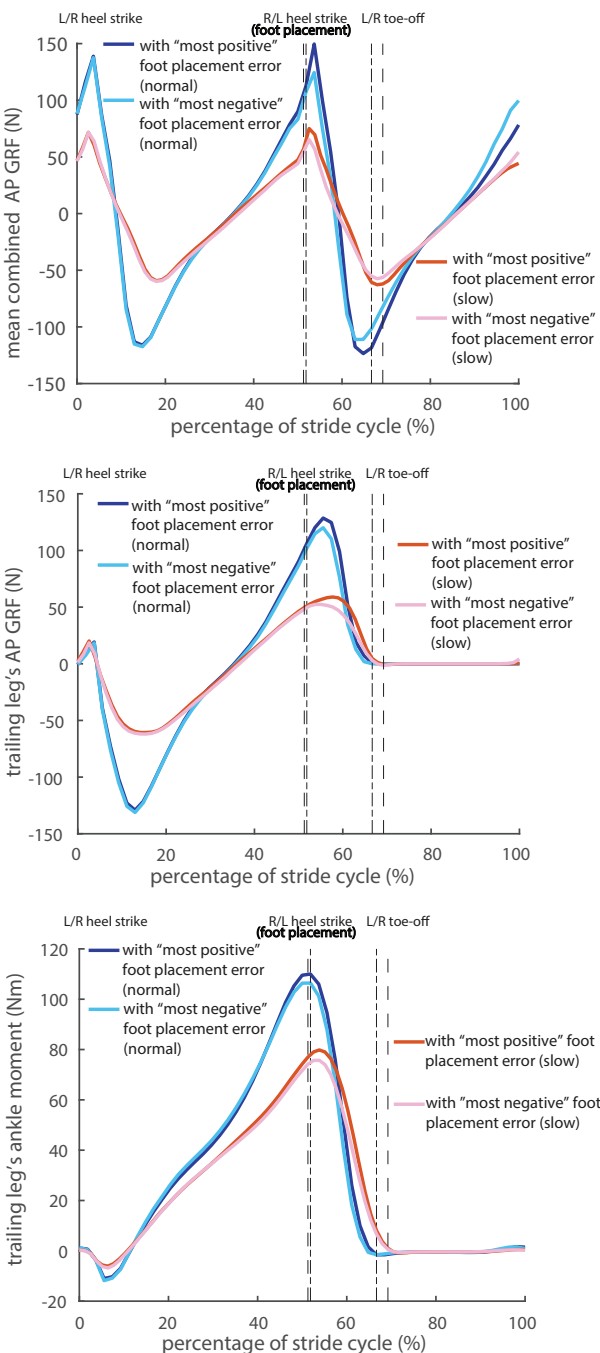

**Figure 5** **Averaged kinetic time series corresponding to the "most positive" and "most negative" foot placement errors in normal and slow walking.** (A) Combined AP GRF; (B) trailing leg's AP GRF; (C) ankle moment. The phases between the dash-dotted lines or the dashed lines correspond to the double stance phase in normal and slow walking, respectively.

strongly determined by the ankle moment. Push-off can be generated by ankle, knee, and hip muscles (*Kuo, 2002*). At the instant of maximal trailing leg power, the gain from the ankle moment to the trailing leg's AP GRF is ten times larger than the gain from the hip joint moment (*Toney & Chang, 2016*). It is therefore not unexpected that correlations for the ankle moment are close to the correlations for the trailing leg's AP GRF.

Peaks in the correlations between foot placement errors and kinetic variables occurred around the middle of the subsequent double stance phase (Figs. 4A–4C), while the peak combined/trailing leg's AP GRF and the peak trailing leg's ankle moment all occurred already in the beginning of the subsequent double stance phase (Fig. 5). Both observations imply that modulations of the CoM states are less prevalent during the peak combined/trailing leg's AP GRF or peak ankle moment, but mostly around the middle of the double stance phase, at which time the CoM AP velocity and ankle power is highest (*Hernández et al., 2009*; *Kuhman & Hurt, 2019*; *Lipfert et al., 2014*). Modulations of the CoM states may be achieved either by adjusting the trailing leg's AP GRF and ankle moment magnitude (*e.g.*, Fig. 5C), or by modulating their timings (duration of the double stance phase and relative duration of particular force/torque in the double stance phase) (*Kuhman & Hurt, 2019*; *Toney & Chang, 2016*; *van Mierlo et al., 2021*; *Williams & Martin, 2019*). For instance, in slow walking the peak in combined AP GRF occurs later in too large foot placement error steps than in too small foot placement error steps (Fig. 5B), which may imply a prolonged activation of the muscles contributing to this force. Yet, the prolonged activation mechanism in slow walking cannot be directly verified or falsified from Figs. 4 and 5 due to the normalization of the double stance phase to a fixed window. The time delay between heel strike and the peak in the correlations between foot placement errors and ankle moment in normal walking was about 80 ms. Feedback delay in control of human ankle moments is arguably longer ($\approx 100$ ms) due to signal transmission, sensory integration in the nervous system and the electromechanical delay (*Afschrift, de Groote & Jonkers, 2021*; *Welch & Ting, 2008*). Rather than directly using foot placement error at heel strike, humans might predict their foot placement error shortly before heel strike as input for the (ankle) push-off modulation. The correlations between the CoM states and the AP foot placement shortly before heel strike were close to the ones at heel strike (Fig. 2). This implies a largely similar foot placement accuracy when predicted prior to or at foot placement. Whether and how humans estimate foot placement error around foot placement, however, requires further investigation.

The positive correlations between foot placement errors and ankle moment at initial double stance phase in normal speed (Fig. 4C) suggest limited ankle push-off modulations before contralateral heel strike. The correlations for ankle moment were positive throughout the double stance phase except at the end, and peak correlations did not appear until the middle of the double stance phase (Fig. 4C). Our findings are consistent with the study by *Kuhman & Hurt (2019)* who found that at human preferred steady-state walking speed (around 1.4 m/s), the peak trailing leg's AP GRF, the onset and peak of trailing leg power and the peak ankle power all occurred after contralateral heel strike. Humans seemingly rely less on the more energy efficient preemptive push-off strategy to stabilize gait than suggested by modeling studies (*Kuo, 2002*; *Ruina, Bertram & Srinivasan, 2005*). Potentially,

they opt for producing push-off somewhat later to incorporate sensory input information about foot placement error.

We showed that steady-state walking relies on more or less the same push-off strategy as perturbed walking, where a forward push led to more forward foot placement (*Vlutters, van Asseldonk & vander Kooij, 2018b*), followed by CoP modulation (shorter double stance duration and longer CoP distance travelled, see *van Mierlo et al., 2021*) from either weight shift of the leading leg (*Hof, 2007*) or plantarflexion ankle moment modulations (*Gruben & Boehm, 2014*; *Vlutters, van Asseldonk & vander Kooij, 2018b*), contributing to a reduction of the CoM AP velocity over the double stance phase. *van Mierlo et al. (2021)* found CoP modulation to be absent for backward perturbations, presumably because adjustments in foot placement sufficiently reduced the foot placement error (see Fig. 5 from *Vlutters, van Asseldonk & vander Kooij, 2018b*). According to *Vlutters, van Asseldonk & van der Kooij (2018a)*, changing magnitude and direction of AP perturbations on the pelvis near foot contact hardly affected the AP foot placement in both the first and second recovery step. This suggests that a limited response time for foot placement modulation primarily calls for push-off modulation. Apparently, humans cope with perturbations by complementing foot placement with push-off control to correct the remaining foot placement errors.

Older adults have poorer foot placement accuracy, at least in the medio-lateral direction (*Arvin et al., 2018*). They have a reduced ankle push-off power generation (*Franz, 2016*; *Hernández et al., 2009*). One may therefore expect the older adults have poorer gait robustness due to larger foot placement errors and a reduced capacity to correct for these. Future work should compare age-related differences in foot placement error at the beginning and the end of the double stance phase and relate them to differences in push-off modulation. This can further clarify gait adaptations in the elderly, such as the distal-to-proximal redistribution of push-off power production (*Franz, 2016*).

We interpreted the modulation of the push-off force by the trailing leg in terms of gait stability. Admittedly, the modulation of push-off force might also be explained by speed regulation. On a treadmill, speed is controlled from step to step and may results in an over-correction of speed errors with respect to the treadmill speed (*Dingwell & Cusumano, 2015*). When walking at different steady-state speeds, humans can adjust the relative duration of the double stance phase (cf. the percentage for normal and slow walking in Fig. 4), which may lead to changes in magnitudes of AP GRF (*Williams & Martin, 2019*). Here, we also observed over-correction of foot placement errors by the combined AP GRF at the subsequent step (see the end of the stride cycle in Figs. 4A and 5A). The foot placement error is determined by two variables: CoM states and foot placement at heel strike. For an inverted pendulum model walking at a nominal speed, perturbations leading to positive foot placement errors reduced walking speed and perturbations leading to negative foot placement errors increased walking speed (*Hof, 2008*). Corrections of foot placement errors probably also correct speed errors with respect to the treadmill. Our current data do not allow for directly distinguishing whether maintaining gait stability or constant speed were the *primary* control goal. Either way, it is likely to be an improper dichotomy given their interdependence (*Hof, 2008*).

Our study suggests that push-off control corrects foot placement errors in steady-state walking. Push-off, however, also contributes to the leg swing (*Zelik & Adamczyk, 2016*) and to the foot trajectory in the early swing phase. This is evident from humans' responses to stepping target perturbations (*Barton, Matthis & Fajen, 2019*). Contrary to push-off control, swing leg control for foot placement is largely passive except at the beginning and end of the swing phase, as indicated by quick bursts of leg swing and retraction impulses (*Doke, Donelan & Kuo, 2005*). This passive ("predictable") pendular dynamics in the swing phase may hence be exploited by the feedforward/anticipative control of push-off for the desired foot placement. It is currently unknown whether in steady-state walking feedforward control of push-off is also used. However, our findings seemed to disagree with the assumption of feedforward control of push-off determining the next foot placement. This can be seen from the fact that (1) the correlations between the foot placement errors and the combined AP GRF at foot placement were close to zero (Fig. 4A), (2) the correlations between the foot placement errors and the trailing leg's AP GRF at the beginning of the subsequent double stance phase were close to zero (Fig. 4B), and (3) the foot placement errors were over-corrected by the combined AP GRF after the subsequent double stance phase (Figs. 4A and 5A). To further unravel the stabilizing mechanisms of push-off control, modeling approaches can elucidate when implementing feedforward and/or feedback push-off controllers based on foot placement errors for simple bipedal walkers under noisy conditions. *Ryu & Kuo (2021)* showed pure feedback and pure feedforward control to be susceptible to sensor noise and process noise (*e.g.*, uncertainty in the environment), respectively, and the combined feedforward and feedback controller was best for stable and robust walking.

## Limitations

Our study comes with several limitations. Before listing them, however, we want to repeat our statement-of-warning, namely, that the correlations we observed should be interpreted with care. Next to feedback control there might be other sources causing them, such as passive dynamics (*Patil, Dingwell & Cusumano, 2019*). Future studies may employ (perceptual or mechanical) perturbations (*Fettrow et al., 2019*; *Roden-Reynolds et al., 2015*) and/or EMG responses (*Rankin, Buffo & Dean, 2014*; *van Leeuwen et al., 2021*; *Vlutters, van Asseldonk & van der Kooij, 2019*) to disentangle contributions from feedback control and from passive dynamics.

We did not apply any normalization, to render our variables of interest easier to interpret. Admittedly, normalization can reduce inter-participant variability. For this, however, we customized each participant's treadmill speed, *i.e.,* we maintained constant dimensionless speeds (*Hof, 1996*). Although this is a valid approach to account for participants' leg differences, controlling speeds and additionally stride frequency by metronome may increase the difficulty of the walking task compared to (unconstrained) natural walking.

We observed an outlier with large negative correlations for the ankle moments only in slow walking in Fig. 4C. This participant's right ankle moments were negatively correlated to (left) foot placement errors, and the foot placement of the right leg was consistently further forward than of the left leg. We suspect this participant used a different and

uncommon walking strategy in slow walking. And there was another participant that might be considered an outlier when considering the correlations between the left trailing leg's AP GRF and right foot placement errors in slow walking in Fig. 4B. However, this participant showed different behavior only prior to heel strike.

## CONCLUSION

We correlated foot placement errors with kinetic time series of ankle push-off to test the idea that foot placement errors are corrected for by push-off in steady-state walking. We found ankle push-off torque to be correlated to AP foot placement errors mainly around the middle of the double stance phase both in steady-state normal and slow walking, confirming our hypothesis that humans employ ankle push-off control to compensate for errors in AP foot placement during steady-state walking.

### Funding

Sjoerd M. Bruijn and Jian Jin are funded by a VIDI grant no. (016.Vidi.178.014) from the Dutch Organization for Scientific Research (NWO). The funders had no role in study design, data collection and analysis, decision to publish, or preparation of the manuscript.

### Grant Disclosures

The following grant information was disclosed by the authors:
VIDI: 016.Vidi.178.014.

### Competing Interests

Jaap H. van Dieën is an Academic Editor for PeerJ.

### Author Contributions

- Jian Jin conceived and designed the experiments, analyzed the data, prepared figures and/or tables, authored or reviewed drafts of the article, and approved the final draft.
- Jaap H. van Dieën analyzed the data, authored or reviewed drafts of the article, and approved the final draft.
- Dinant Kistemaker analyzed the data, authored or reviewed drafts of the article, and approved the final draft.
- Andreas Daffertshofer analyzed the data, authored or reviewed drafts of the article, and approved the final draft.
- Sjoerd M. Bruijn conceived and designed the experiments, analyzed the data, prepared figures and/or tables, authored or reviewed drafts of the article, and approved the final draft.

### Data Availability

   The data and code for the analyses are available at Zenodo: Jin, Jian, van Dieën, Jaap, Kistemaker, Dinant, Daffertshofer, Andreas, & Bruijn, Sjoerd. (2022). Code

and data for manuscript: Does ankle push-off correct for errors in anterior-posterior foot placement relative to center-of-mass states? (preprint) [Data set]. Zenodo. https://doi.org/10.5281/zenodo.7393791.

## Supplemental Information

Supplemental information for this article can be found online at http://dx.doi.org/10.7717/peerj.15375#supplemental-information.

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
