# Peer review of "Does ankle push-off correct for errors in anterior–posterior foot placement relative to center-of-mass states?"

_PeerJ, doi:10.7717/peerj.15375_

## Round 0.1 · original submission · Minor Revisions

Please, consider the comments of all reviewers. They will improve the manuscript. Thank you.

·

Basic reporting

This manuscript re-analyzes an existing data set to investigate whether humans modulate push-off as a feedback control mechanism to maintain balance in the anterior-posterior direction during walking. Previous studies indicate that foot placement is actively used for balance control. The hypothesis here is that humans use the foot placement error as input for a feedback control law and modulate their push-off force accordingly, to correct for foot placement error. This predicts that foot placement would be correlated with kinetic variables like ground reaction force and ankle moments. The authors indeed find these expected correlations in the data, supporting the hypothesis.

This is a valid research question and a well designed analysis. The manuscript is largely well written. I have only one slightly-larger-than-minor and several minor issues that I hope the authors can address before publication.

Slightly-larger-than-minor issue: Correlations are a central aspect of the methods, but some details are not specified. What is the “combined AP GRF” (I assume combined from both legs, but please clarify). I assume that GRF and foot locations use the same coordinates? Then positive correlation would mean that more anterior foot placement error relates to larger forward GRF, as predicted by the feedback control hypothesis, is that right? These relationships are not obvious, and it might be good to spell them out, rather than leaving them for the reader to think through themselves. What are the directions of the ankle moments, i.e. is flexion moment positive or negative? Please specify that and describe the expected correlation predicted from the feedback control hypothesis here as well.

Minor issues
l. 46: “Foot placement depends on position and velocity of the center-of-mass (CoM) and push-off modulates with deviations between actual and predicted CoM trajectories.” — the second part of this reads like it’s a fact, rather than a control law. Maybe replace “modulates” with “covaries”?

l. 50 “On the one hand, the covariance between CoM states and anterior-posterior foot placement served as a measure of foot placement control” — this is hard to understand for non-experts, I suggest elaborating slightly on this point

l.118 “Since CoM states covary with AP foot placement, neither of them can be isolated as input to push-off control” — cannot be isolated by whom? For an observer, one variable is pretty much as good as another, and isolating either shouldn’t be a problem and is what you do in your analysis. For the nervous system, by “isolating” you might mean something like “can be sensed or estimated”, which is required for a variable to be used as input for a feedback control law, and this seems to be what you imply here. If so, do you imply that the foot placement error does not have the same issue? I would question that, since the foot placement error seems to be substantially harder to estimate from the available sensory information than the CoM state. But I’m speculating here, please clarify what you mean here and, if applicable, maybe discuss my speculative points.

l.123 “subserve” — not sure what that term means here, please clarify

l. 127 “deviations from this covariance” — what is a deviation from a covariance? Do you mean Do you mean from the regression line?

l. 171 “The imposed frequency was customized as the average preferred stride frequency during the last hundred steps of the familiarization trial at each speed.” — that seems to imply that there was no metronome during the familiarization, but the metronome was introduced on the first actual trial, so there was no familiarization to walking with a metronome, is that correct? Please clarify.

l. 188 “a left (right) step [was defined] from left (right) toe-off to right (left) toe-off”, but l.206: “the index indicates the time instant in a step (0-100% from heel strike to contralateral heel strike)” — in one definition a step is from toe-off to toe-off and in the other from heel-strike to heel-strike. Please clarify.

Equation 2: the index “i” here seems to indicate the step within the trial. In l.206 above, however, you specify that “i” indicates the time instant in a step. This doesn’t seem to make sense in Equation 2. Please clarify.

l.249 “The distributions of foot placement errors across all participants for normal and slow walking were close to Gaussian with zero mean” — maybe add that the “zero mean” is necessary from the definition of the variable and not an observation

l.308 “we found maximum correlations” — what exactly is maximal and within what set? (Also maximum —> maximal)

Spelling and grammar
l. 242 coefficient —> coefficients

l.331 remove period after “Figures”

Experimental design

This is a re-analysis of an existing data set. The data set is suitable to answer the research question.

Validity of the findings

no comment

·

Basic reporting

No comment

Experimental design

No comment

Validity of the findings

No comment

Additional comments

Authors presented a very well-organized manuscript with clear explanation. I would like to thank them for their contribution and for their efforts. Their contribution is clear and justified with their findings and I have very minor comments as follows:
- Line 54: I think authors meant "indicator" in this sentence.
- Line 134: It would be better to specify the push-off side (leading or trailing or both) here to avoid confusion.
- Line 185 and 188: Authors mention about ankle moments determination with two references in separate places. Please clarify and justify.
- Eqn (1): Could you clarify whether the position and velocity here are vectors or not? If not then it would be better to redefine them, since position and velocity are vectors. So this equation needs clearer explanation.
- Line 206: Authors defined a step from toe-off to contralateral toe-off but here it is defined from heel-strike to contralateral heel-strike. Please clarify and justify.
- Fig. 4: It would be better to use different color than red for improving the visibility of mean curve. Please also revise (b) with dash-dotted line, since both are dashed in this one.
- When I looked at Fig. 5, I just wondered why authors did not include trailing leg ankle moment into their evaluations? This could be clarified and further discussed in the discussion.
- Line 331: "." after Figures should be omitted.
- Line 406: "uncertainty"

Reviewer 3 ·

Basic reporting

I would like to thank the editor and the authors for giving me an opportunity to review this interesting and well-written manuscript. In this manuscript, the authors examined the relationships between foot placement control, ankle push-off torque, and anterior-posterior center-of-mass trajectories. The strengths of this manuscript are: 1) the rationale and motivations for the study, which are well-justified through prior modeling and experimental studies examining the control mechanisms during walking, and 2) the unique analyses that are aimed to examine the mechanisms of gait control. If there is a weakness, it is in the protocol design (e.g., use of a metronome) and the clarity of the description of the correlation analyses. My comments below are ordered by the importance of the issues.

1. Ln 170-172: I have a few questions and concerns regarding the use of a metronome. As one of the purposes of this project is to understand the natural step-to-step variations in foot placement, which are directly linked to stride lengths, I’m wondering why the metronome was used. When walking at a fixed speed on a treadmill, it seems like minimizing the stride frequency variations (via metronome) would also minimize the stride length variations, and hence anterior-posterior foot placement. I wonder if this protocol would over-constrain the subjects’ variations and make it more difficult to assess the natural variability. In other words, I wonder if the correlations that are reported here could potentially be higher (or lower) if the correlations were assessed during a less-constrained walking task.
I’m also wondering whether the instructions to match the right heel strikes to the beat of the metronome would present inter-limb biases. I understood that the foot placement models were independent of left versus right limbs. However, I wonder if the foot placement error corrections were stronger on one limb over the other – for example, did subjects have a tendency to rely more on the right step (or left ankle moment) for error correction since the metronome feedback was provided for the right heel strike?

2. In general, I got confused in multiple places about whether the correlations were done for one variable relative to another variable during the same step or the subsequent step. I’ve noted specific places where further clarifications may be helpful.
• LN 211-213: I presume that this ‘during the double stance’ is for the next step, and not the same step as the ‘foot placement error at heel strike’. Please clarify. A very similar comment in results/discussion sections (e.g., see LN 257-259)
• LN 214: I understood that the (i+1) signify the step-phase in the next step. However, if ‘I’ goes from 0 to 100, then ‘i+1’ would still be in the same step. Should this be ‘i+100’?

3. Eq 1 and 2 do not show any of the kinetic variables that were also included in the regression models. Should those variables also be included? Also, having a clear description of the regression models that include the kinetic variables would be helpful. Related to my comments above (see #2 above), were the kinetic variables used to assess their impact on the foot placement errors on the same step or the subsequent step? I would presume the latter, but further clarifying this throughout the Methods and Results/Discussion (e.g., LN 257-259, LN 268-269, LN 278-280, LN 296-298, LN 308-310, LN 317-318 etc) would be helpful for the readers. An example of a sentence that is easier to understand is LN 304-305: using a sentence like this throughout the manuscript would be helpful to the readers.

4. Ln 132-136: The rationale here is presented very clearly. However, based on what is provided here, I’m wondering why the authors decided to focus their analysis on ankle torque and not ankle work. These cited studies seem to motivate analyzing ankle work.

5. I find the discussion regarding feedforward versus feedback very insightful (e.g., LNS 392-407). “As such it remains opaque whether push-off corrects foot placement error or push-off plans for the next foot placement”. If the latter is true, could push-off cause the foot placement error? It seems like this possibility was refuted by the authors (see LN 263-264, 272-274), but it may be helpful to clarify this point.

6. Ln 206: Here, a step cycle is defined from heel strike to contralateral heel strike, whereas Figure 2 shows a step cycle defined as toe-off to contralateral toe-off. Keeping them consistent is advised.

Experimental design

see my comment #1 above.

Validity of the findings

no comment

Additional comments

This was a fun manuscript to read. I look forward to seeing the revisions.

---

## Round 0.2 · accepted · Accept

The authors have addressed all reviewers' comments. The manuscript is ready for publication. Please, consider the minor revisions of Reviewer 3 in the final review phase prior to the publication of the manuscript. Congratulations!

·

Basic reporting

The authors addressed all my concerns satisfactorily. Congratulations on a well-done study and a well-written manuscript.

Experimental design

N/A

Validity of the findings

N/A

Additional comments

N/A

·

Basic reporting

No comment

Experimental design

No comment

Validity of the findings

No comment

Additional comments

I would like to thank authors for addressing all of my comments and for their efforts to improve the manuscript. I have no further comments.

Reviewer 3 ·

Basic reporting

no additional comments

Experimental design

no additional comments

Validity of the findings

no additional comments.

Additional comments

The authors did a great job of addressing the reviewers' comments. I have two very minor points that the authors can easily fix.
1. Abstract Line 46: there should be a period after 'ankle push-off control' (and not a colon).
2. Line 147: work and impulse are kinetic variables (and not kinematic).